# Influence of Diet Enriched with Cocoa Bean Extracts on Physiological Indices of Laboratory Rats

**DOI:** 10.3390/molecules24050825

**Published:** 2019-02-26

**Authors:** Dorota Żyżelewicz, Małgorzata Bojczuk, Grażyna Budryn, Adam Jurgoński, Zenon Zduńczyk, Jerzy Juśkiewicz, Joanna Oracz

**Affiliations:** 1Institute of Food Technology and Analysis, Faculty of Biotechnology and Food Sciences, Lodz University of Technology, 90-924 Lodz, Poland; malgorzata.bojczuk@gmail.com (M.B.); grazyna.budryn@p.lodz.pl (G.B.); joanna.oracz@p.lodz.pl (J.O.); 2Department of Biological Functions of Food, Institute of Animal Reproduction and Food Research, Polish Academy of Sciences, 10-748 Olsztyn, Poland; a.jurgonski@pan.olsztyn.pl (A.J.); z.zdunczyk@pan.olsztyn.pl (Z.Z.); j.juskiewicz@pan.olsztyn.pl (J.J.)

**Keywords:** cocoa bean extracts, flavonoids, metabolism indicators

## Abstract

Cocoa bean is a rich source of polyphenols, mainly flavonoids which have a wide range of biological properties. The aim of the study was to determine the physiological indices of laboratory rats as a response to diets containing water extracts of raw or roasted cocoa beans of *Forastero* variety, as well as purified monomeric flavan-3-ols fraction isolated from them. The influence of these extracts on selected parameters was studied during 4 weeks feeding. The samples of rats feces were collected throughout the experiment and after its completion, biological samples (intestines content, blood, and organs) were retrieved individually from each rat and subjected to analyses. The observed changes in the gastrointestinal tract functioning indices and metabolism indicators, determined throughout the study and after its completion, confirm to some extent the biological activity of polyphenol extracts of cocoa beans. The differences in the results obtained for the analyzed parameters of the gastrointestinal tract revealed that the cocoa bean extracts differently affected the physicochemical properties of rats’ intestines. The results indicate the beneficial effects of the applied nutrition treatment on the activity of cecal enzymes and the content of volatile fatty acids in the gut.

## 1. Introduction

Plant tissues have developed antioxidant systems to control free radicals, lipid oxidation catalysts, oxidation intermediates, and secondary breakdown products [1,2,3]. They are the richest natural source of bioactive compounds—polyphenols known for their antioxidant properties. These compounds include flavonoids, phenolic acids, carotenoids, and tocopherols that can inhibit Fe^3+^/AA-induced oxidation, scavenge free radicals, and act as reductants [4,5]. Phenolic compounds are widely found in a variety of fruits, nuts, seeds, and cereals [6]. Other sources include chocolate and cocoa beans which are indeed reported to contain more phenolic compounds and have higher antioxidant activity than such beverages as tea or red wine [7].

Cocoa beans are seeds of the tropical tree *Theobroma cacao* L. The three main varieties include *Forastero*, *Criollo*, and *Trinitario*. The varieties exhibit differences in the appearance of pods, yields of beans, flavor characteristics and in resistance to pests and disease [8,9]. The *Forastero* variety is the most common and widely cultivated for mass production. It is characterized by more robust and sour-bitter in flavor seeds, however, seeds of this variety are also richer in cocoa butter and have a higher content of phenolic compounds compared to the seeds of other varieties [10]. Although a practical use of cocoa bean has originated from ancient civilizations in South America (Olmecs, Mayas, and Aztecs), it gains more and more recognition in confectionery, cosmetic and pharmaceutical industries [11,12,13]. They contain a significant amount of bioactive compounds (mainly epicatechin, catechin, and procyanidins) which allows them to exhibit a wide range of physiological properties such as antioxidant, antiradical, antimicrobial, anti-inflammatory, antiatherogenic, antithrombotic, antihypertensive, anticarcinogenic, and cardioprotective resulting in the protection against diseases among others such as coronary heart disease, cancer or neurodegenerative disorders [14,15,16,17,18,19,20,21,22,23]. 

Cocoa is largely produced in developing countries but is mostly exported to and consumed in industrialized countries. Taking into account the volume of exports, the two main cocoa producing countries are Ivory Coast and Ghana with an average of around 40% and 20% of the annual global production, respectively. They are followed by Indonesia, Nigeria, Cameroon, Brazil, Ecuador, Dominican Republic and Malaysia, and together supply about 90% of the world production [24,25]. 

After harvesting, cocoa beans undergo several pre-processing steps such as breaking, fermentation, and drying. They are quite important in ensuring high quality of the final product. The crucial step for the formation of the flavor, however, is the roasting. It is the most important technological operation throughout the whole processing, and the nature of chemical and physical changes (also in the qualitative and quantitative composition of polyphenols) depends on the number of parameters of this process [26,27,28,29,30]. Cocoa beans that are not roasted are characterized by a bitter, acidic, astringent, and nutty flavor. Roasting reduces the concentration of volatile acids which leads to the decrease in the beans’ acidity [8,31]. 

The aim of this study was to check the effects of water extracts from raw and roasted cocoa beans of *Forastero* variety and a monomeric flavan-3-ols fraction isolated from them on the physiological indices of gastrointestinal tract and hematological parameters in rats fed a high-fat low-fiber diet. These are preliminary studies focused on the early stages of a high fat diet intake. 

## 2. Results and Discussion

As shown in Table 1, the enrichment of diets with cocoa bean extracts and monomeric flavan-3-ols fraction varied the values of the analyzed parameters of rats’ small intestine, cecum, and colon. The presence of the freeze-dried cocoa bean extracts increased the content of the small intestine, probably as a result of slowing down the transit of the content. A statistically significant difference was recorded between D_RT_ group and both D_CT_ and D_CF_ groups. There were no differences (*p* ≥ 0.05) in the pH of the small intestine contents in all experimental diets. In relation to the D_CF_ diet, there were no differences in the sucrase activity in the intestinal epithelium of rats receiving diets containing cocoa bean extracts. The maltase and lactase activities were higher in the D_RW_ group and the lowest for the D_CF_ group. The other diets containing the extracts resulted in similar maltase activity. The high-fat diet reduced the activity of maltase, while the presence of the cocoa bean extracts in other diets eliminated this effect. The mass of the wall and contents of the rats’ cecum differed between experimental diets, however, statistically significant differences (*p* ≤ 0.05) were recorded only in some cases (Table 1), i.e., in the case of the tissue mass, D_CS_ and D_RT_ groups the value is higher than in the D_CT_ group. D_CT_ and D_RW_ group differed in the content of dry matter in the cecum, whereas D_CF_ and D_RT_ contained significantly different amounts of the ammonia. The diet did not differentiate the weight of tissue and content of colon, which indicates that no changes caused by diets were noted during the experiment. In this part of the intestine, the changes could indicate an effect on the microbiota growth. 

Nevertheless, the applied experimental diets diversified the activity of fecal microbiota enzymes (Table 2). In relation to the first measurement of β-glucosidase activity (before the experiment), a very high and varied increase in the activity of this enzyme after the introduction of the experimental diets was observed. After the first and second week, the results of the β-glucosidase activity were more even though higher (especially in the second week) in the case of diets containing extracts as compared to both control groups. A similar tendency was observed in the case β-galactosidase activity, i.e., a high increase at the beginning of the experiment and a decrease after the period of adaptation to the administered diets. In the case of β-galactosidase, a significant activity increase was caused by the DCF diet and the influence of cocoa bean extracts was not statistically confirmed. After 2 weeks of the experiment, significantly higher β-glucuronidase activity was observed in the feces of rats fed D_RW_ and D_RT_ diets, whereas the result in the D_CT_ group did not differ statistically from the ones determined for both control groups.

At the end of the experiment, the applied experimental diets diversified the activity of rats’ cecum microflora enzymes (Table 3). In comparison to the standard one, the high-fat diet reduced the total α-glucosidase activity of the microflora in the rats’ cecum. The presence of raw and roasted cocoa bean extracts increased the activity of this enzyme, whereas monomeric flavan-3-ols fraction showed no effect. Compared to D_CS_, the high-fat control diet reduced the β-glycosidase activity in the contents of rats’ cecum, while extracts from cocoa beans maintained the activity of this enzyme at the level of the control standard diet. A large variation in α-galactosidase activity was recorded, with the D_RT_ group expressing the highest and D_CT_ the lowest activity of this enzyme among all experimental diets. The high-fat diet increased the β-galactosidase activity in the rats’ cecum as compared to the D_CS_ diet, and only the extract from roasted cocoa beans reduced the activity of this enzyme to the level of the group fed control standard diet. No significant differences (*p* ≥ 0.05) in the microbial activity of β-glucuronidase in the content of the rats’ cecum were determined.

The applied experimental diets resulted also in a diversification in the concentration and total production (pool) of volatile fatty acids (VFAs) in the cecum content (Table 4). VFAs are products of bacterial fermentation in the distal intestine of rats and other omnivores. The three major are propionate, acetate, and butyrate. It is important to measure the utilization of VFAs since their proportions can be manipulated by dietary means [32]. VFAs are known to play a significant role in maintaining colonic physiology. Butyrate is particularly important for normal development of colonic epithelial cells due to its anti-carcinogenic and anti-inflammatory properties. It also constitutes the primary and at the same time preferred energy source for colonocytes [33,34]. The results obtained reveal a small impact of the applied nutritional measures on the concentration of acetic acid, the predominant VFA in the rats’ cecum. The D_RW_ group tend to had higher acetic in the cecal content than the other groups (*p* = 0.052). There was a differentiation in the concentration of individual acids, whereas the sum of VFAs in the intestinal content did not differ (*p* ≥ 0.05) between the groups. The total production (pool) of VFAs in the cecum content was only slightly different between diets, however, a statistically significant difference (*p* ≤ 0.05) was noted only in the case of a diet containing monomeric flavan-3-ols fraction. The VFAs pool was significantly differentiated (*p* ≤ 0.05) between individuals in the same group and for this reason, quite large differences between groups were not statistically confirmed. Compared to D_CS_ group, a significantly higher production of acetic acid was found in all other groups. Different results were obtained for butyric acid, whose concentration decreased significantly in D_RW_, D_RT_, and D_CT_ groups.

Hematological parameters in rats are shown in Table 5. Most of the values differed to some extent among the tested groups; however, at the same time, they fitted in our laboratory norms (details below Table 5). An exception were intermediate forms between lymphocytes and neutrophils (MID), whose percentage was elevated in both control groups (D_CS_ and D_CF_), whereas all tested extracts slightly or significantly decreased their value (group D**_CT_** or groups D**_RW_** and D**_RT_**, respectively, vs. group D_CF_). The hemoglobin concentration (HGB) was in turn below the norm, but only in the D_CS_ group, whereas the platelet count was below the norm in both control groups and in the D**_CT_** group. 

## 3. Materials and Methods 

### 3.1. Chemicals and Reagents

Ethyl acetate and n-hexane were obtained from CHEMPUR (Piekary Śląskie, Poland). All other reagents used were purchased from POCH (Gliwice, Poland). Ultrapure water was obtained from a Millipore Milli-Q Plus purification system (Bedford, MA, USA).

### 3.2. Materials

The basic material used in the following study was raw and roasted cocoa bean (*T. cacao* L.) of *Forastero* variety harvested in Peru. It was purchased from Barry Callebaut Polska Ltd. (Lodz, Poland). Raw and roasted cocoa beans were used to prepare freeze-dried water extracts. Roasting process parameters are described further in this section. Additionally, the fraction of monomeric flavan-3-ols was separated from cocoa bean extracts by countercurrent partition chromatography (CPC).

#### Cocoa Bean Processing

##### Roasting

A batch of cocoa beans was convectively roasted in the tunnel roaster according to the method described by Żyżelewicz et al. (2014) [35]. The process was conducted for 35 min at 135 °C ensuring the water content in the final product around 2%. The parameters of the roasting air were 1 m/s flow velocity and 0.3% relative humidity.

##### Preparation of Cocoa Bean Extracts (CBEs)

CBEs were prepared in accordance with the procedure described by Żyżelewicz et al. (2016) [36]. Raw and roasted cocoa beans were dehulled, ground and sieved to a particle size ranging from 0.200 to 1.00 mm. This was done to achieve satisfactory extractability of phenolic compounds from beans and effective filtration. The extracts were prepared using distilled water in 1:3 (*w*/*w*) ratio, for ground cocoa beans and water, respectively. In this study, water was chosen as a solvent due to the fact that the presented study is a part of a greater research which ultimate goal is to apply obtained CBEs as well as isolated from them fractions in the food industry. The extraction of phenolic compounds from raw and roasted cocoa beans was carried out at 60 °C for 30 min using an SV 1422 Memmert (Schwabach, Germany) water bath with shaking. After this time, the suspensions were filtered under vacuum using a vacuum pump KNF 18 035.3 N (Neuberger, NJ, USA). Finally, part of the obtained extracts was frozen, freeze-dried in a BETTA2-8LSC plus Christ freeze drier (Osterode am Harz, Germany) and stored at −24 °C until further usage. The total phenolics concentration in freeze-dried extracts was 32.48 mg/g DW and 28.62 mg/g DW, respectively for freeze-dried CBE from raw and roasted cocoa beans. A UHPLC-DAD-ESI-MS/MS analysis was used to quantify and to identify individual polyphenols and methylxanthines in the obtained extracts. (+)-Catechin (0.24 ± 0.01 mg/g DW), (−)-epicatechin (8.68 ± 0.04 mg/g DW), epigallocatechin (1.36 ± 0.01 mg/g DW), procyanidin B2 (7.59 ± 0.02 mg/g DW), procyanidin C1 (2.44 ± 0.02 mg/g DW), other procyanidins (11.40 ± 0.05 mg/g DW), quercetin (0.01 ± 0.00 mg/g DW), quercetin 3-*O*-glucoside (0.30 ± 0.01 mg/g DW), quercetin 3-*O*-arabinoside (0.31 ± 0.01 mg/g DW), and quercetin 3-*O*-galactoside (0.09 ± 0.01 mg/g DW) were present in the raw cocoa beans extract. Roasted cocoa beans extract contained (+)-catechin (0.47 ± 0.02 mg/g DW), (−)-epicatechin (6.35 ± 0.03 mg/g DW), epigallocatechin (0.34 ± 0.01 mg/g DW), procyanidin B2 (7.21 ± 0.02 mg/g DW), procyanidin C1 (1.36 ± 0.01 mg/g DW), other procyanidins (10.41 ± 0.04 mg/g DW), quercetin 3-*O*-glucoside (0.26 ± 0.01 mg/g DW), quercetin 3-*O*-arabinoside (0.27 ± 0.01 mg/g DW), quercetin 3-*O*-galactoside (0.08 ± 0.00 mg/g DW), and gallic acid (1.85 ± 0.02 mg/g DW) [36]. Concentrations of theobromine in extracts from raw and roasted cocoa beans were 57.56 ± 0.23 mg/g DW and 57.69 ± 0.19 mg/g DW while the concentrations of caffeine were 5.03 ± 0.12 mg/g DW and 5.65 ± 0.10 mg/g DW respectively. 

Another part of the obtained extract was subjected to fractionation and purification by CPC. This technique operates based on the principles of a two-phase liquid-liquid partitioning chromatography [37]. The process is described in more details in the next section of this article. 

##### Separation and Purification of Cocoa Bioactive Compounds

The process of separation and purification of the part of the obtained CBE was carried out using CPC technique with the aid of chromatograph SPOT Prep II 50 from Armen Instrument (Saint-Avé, France) integrated with a 2-channel UV/VIS detector, a fraction collector and additional external pump. The apparatus was equipped with Armen Glider CPC v5.0b.11 software. The two-phase solvent system was prepared according to Delaunay, Castagnino, Chèze and Vercauteren (2002) [38]. It was composed of four solvents: hexane, ethyl acetate, ethanol, and water in the ratio of 1:8:2:7 (*v*/*v*/*v*/*v*), respectively. The subsequent steps of CPC complete cycle, as well as details concerning fractions concentration and purification, are described by Żyżelewicz et al. (2016) [36]. The separation of bioactive compounds present in cocoa beans resulted in obtaining three fractions of these compounds: monomeric flavan-3-ols, procyanidins, and colored compounds. The fraction of interest for this study was the first obtained, namely the monomeric flavan-3-ols fraction, rich in catechins. Upon obtaining it was subjected to freeze-drying in a BETTA2-8LSC plus Christ freeze drier (Osterode am Harz, Germany) and stored at −24 °C until further usage. The total content of polyphenols in the monomeric flavan-3-ols fraction was 685.12 mg/g DW. It contained (+)-catechin (27.08 ± 0.03 mg/g DW), (−)-epicatechin (575.57 ± 0.11 mg/g DW), quercetin (0.33 ± 0.02 mg/g DW), quercetin 3-*O*-glucoside (4.07 ± 0.02 mg/g DW), quercetin 3-*O*-arabinoside (4.60 ± 0.01 mg/g DW), quercetin 3-*O*-galactoside (1.49 ± 0.01 mg/g DW) and gallic acid (14.63 ± 0.04 mg/g DW) [36]. In the monomeric flavan-3-ols fraction we also detected a small amounts of theobromine (7.40 ± 0.11 mg/g DW) and caffeine (0.15 ± 0.02 mg/g DW).

### 3.3. Animal Study

#### 3.3.1. General Information on the Tested Subject and Investigated Diets

The study was conducted in compliance with the European guidelines for the care and use of laboratory animals (The ethic approval number: 71/2014), according to the proposal approved by the Local Institutional Animal Care and Use Committee (Olsztyn, Poland). The experiment was conducted on 40 male Wistar laboratory rats with an initial age of four weeks and similar initial body weight. The animals were randomly divided into 5 groups, 8 rats each. The experiment lasted 4 weeks. During this time, the animals were kept individually in plastic cages under the conditions of stable temperature (21–22 °C) and relative humidity of 50–70%, with a ventilation rate of 15 air changes per hour and 12-hour light/12-hour dark cycle. Throughout the whole experiment, the rats had free access to water and food. The experimental diets were a modification of semi-synthetic AIN-93G diet [39], developed by American Institute of Nutrition for rats during their intensive growth and differed in the composition of the added polyphenol extract. The source of protein in the diet was casein, and the sources of minerals and vitamins were standard mineral and vitamin blends, respectively AIN-93G-MX and AIN-93G-VX [40]. In all diets, the DL-methionine was added in order to supplement the deficiency of this amino acid in casein. The detailed composition of all experimental diets is presented in Table 6. Previous studies have demonstrated that in very high doses, flavonoids can act as pro-oxidant [41]. Therefore, due to the fact that the fraction of monomeric flavan-3-ols have an approximately 20-fold higher polyphenols content than extracts of raw and roasted cocoa beans, various additives of the tested preparations were applied to the diets so as to obtain a similar level of total polyphenols in the diet. The rats were fed five types of diets, two control ones (D_CS_ and D_CF_) and three supplemented, with freeze-dried raw (D_RW_) and roasted cocoa beans extracts (D_RT_) as well as with the monomeric flavan-3-ols fraction (D_CT_). D_CS_ diet was an example of a standard one, with a standard composition providing adequate levels of dietary fiber (5% cellulose) and appropriate energy ratios from fat (7% rapeseed oil) and easily digestible carbohydrates (10% sucrose and 53% corn starch). D_CF_ diet, on the other hand, was an example of a high-fat, low-fibre diet referring to the dietary habits of a significant part of the population of economically developed countries. This ‘faulty’ diet was enriched with raw and roasted cocoa bean extracts and monomeric flavan-3-ols fraction, with the aim of checking the possibility of limiting the physiological effects of such type of diet as compared to the standard one.

The investigated diets were administered during the period of intensive growth of tested animals, i.e., from ages of 4 to 8 weeks, with feed intake controlled on a daily basis.

#### 3.3.2. Sample Collection and Analysis

The rats’ feces samples were collected throughout the experiment. After the termination of the study, i.e., after 4 weeks of the experimental feeding, the rats were weighed and anesthetized with sodium pentobarbital (50 mg/kg body weight) according to Close et al. (1997) [42]. The samples of intestinal contents and tissue samples from rats’ organs were collected post-mortem.

Prepared small intestine, cecum, and colon with contents were weighed and pH of the content was determined using microelectrodes (pH-meter, model 301, Hanna Instruments, Amorim - Póvoa de Varzim, Portugal). The ammonia content in the cecum samples was determined according to Hofirek and Haas (2001) [43]. 

The profile of VFAs in the cecum content was determined using a gas chromatograph equipped with FID detector (Shimadzu GC-14A, Kyoto, Japan). The separation was carried out using a 250 mm × 2.6 mm glass chromatography column containing 10% SP-1200/1% H_3_PO_4_ for 80/100 Chromosorb W AW. The column temperature was 110 °C, the injection temperature of 195 °C. The content of the cecum (0.2 g) was mixed with 0.2 mL of methane acid. Next, the sample was diluted with deionized water, centrifuged at 10,000 rpm for 5 min and the supernatant was applied to the chromatography column. 

The activity of glycolytic enzymes was determined according to the method described by Djouzi and Andrieux (1997) [44] modified by Juśkiewicz and Zduńczyk (2002) [45] and Jarosławska et al. (2011) [46]. The β-glucuronidase, α- and β-galactosidase, and α- and β-glucosidase activities were measured colorimetrically indicating the amount of *p*- or *o*-nitrophenol released from the substrates. The solution of the intestinal content was obtained after a disintegration of the appropriate amount of cecum content in 0.1 M phosphate buffer (pH 7.0) and after centrifugation at 13,000 rpm. The color reaction was observed after combining 0.3 mL of the substrate solution (5 mM) and 0.2 mL of the solution of the intestinal content (1:10 *v*/*v*) in 0.1 M phosphate buffer (pH 7.0). After 10 min of incubation at 37 °C, the reaction was terminated by adding 2.5 mL of 0.25 M sodium carbonate. The concentration of *p*-nitrophenol was measured at λ = 400 nm and *o*-nitrophenol at λ = 420 nm. The activity of the enzyme (U) in 1 g of the cecum content was expressed in μmol *p*-(*o*-) nitrophenol released in 1 min. 

To measure the activity of enzymes present in rats’ gastrointestinal tract such as α-glucosidase (maltase), β-fructosidase (sucrase), and β-galactosidase (lactase), mucosal samples were taken from the same fragment of the rats’ small intestine. For this purpose, after its dissection, the small intestine was divided into four equal parts. The second part (the jejunum), was rinsed with cold saline, dissected along a cooled glass plate and then a mucose was collected using a microscope slide. After homogenization of the mucosa in the saline solution, the mixture was centrifuged (9500× *g*, 10 min, 4 °C) and the activity of the above-mentioned enzymes was measured using a modified method of Dahlqvist (1964) [47]. The solutions of maltose, sucrose, and lactose (0.1 mol/L) served as the substrates for the analysis. The enzymatic reaction was carried out for 15 min at 37 °C, after which time the enzymes were deactivated at 100 °C for 3 another minutes. The concentration of the resulting reaction product (glucose) was measured using a reagent kit (Alpha Diagnostics Ltd., Warsaw, Poland). The enzymatic activity was expressed in μmol of glucose released in 1 min, calculated per 1 g of protein present in the mucosa. The protein concentration was measured using the Bradford reagent. 

All haematological parameters were measured in freshly drawn whole blood samples supplemented with heparin using the Abacus Junior Vet analyser (Diatron, Budapest, Hungary). 

#### 3.3.3. Statistical Analysis

Analyses were carried out in triplicate. All obtained results were subjected to statistical analysis using STATISTICA 10 software (StatSoft Inc., Tulsa, OK, USA). The determination comprised of both, average values and one-way analysis of variance ANOVA Tukey’s honest significant difference post hoc test at the significance level of *p* ≤ 0.05.

## 4. Conclusions

In the present study, laboratory rats were fed five different experimental diets differing in their composition. The study aimed at confirming the beneficial effect of food supplementation with the polyphenol-rich material i.e., extracts of raw and roasted cocoa beans as well as isolated from them a fraction of monomeric flavan-3-ols (mainly catechins). Health-promoting benefits of cocoa beans have been reported by numerous studies over the last years. The observed changes in the gastrointestinal tract functioning and metabolism indicators, determined throughout the study, indicate on the biological activity of polyphenol extracts and other components of cocoa beans present in the prepared extracts. The beneficial effect of diet supplemented with cocoa extracts was observed mostly as decreased activity of glycosidases and increased content of volatile fatty acids in cecum. Nevertheless, a relatively short period of diet administration, namely 4 weeks, was not fully sufficient to clearly confirm course of the changes observed within tested groups and the high-fat low-fiber diet used in this study was not able to disturb hematological parameters. Therefore, the potential chemoprotective effect of cocoa extracts could not be observed and the material is recommended for the further analysis.

## Figures and Tables

**Table 1 molecules-24-00825-t001:** Parameters of the small intestine, cecum, and colon of rats fed experimental diets. Data are expressed as mean ± SEM, *n* = 8.

	Diet					*p*-Value
	D_CS_	D_CF_	D_RW_	D_RT_	D_CT_	
**Small intestine**						
**Weight with content ^1^**	1.98 ± 0.09 ^a^	2.10 ± 0.11 ^b^	2.16 ± 0.12 ^ab^	2.30 ± 0.10 ^a^	2.05 ± 0.09 ^b^	0.002
**Content pH**	7.33 ± 0.06	7.18 ± 0.04	6.92 ± 0.06	7.32 ± 0.05	7.17 ± 0.04	0.053
**Sucrase activity ^2^**	6.53 ± 0.47 ^a^	3.69 ± 0.39 ^b^	4.83 ± 0.39 ^b^	3.97 ± 0.47 ^b^	4.39 ± 0.48 ^b^	0.004
**Maltase activity ^2^**	35.8 ± 0.78 ^a^	22.6 ± 0.86 ^b^	35.4 ± 0.82 ^a^	27.7 ± 0.68 ^ab^	28.5 ± 0.71 ^ab^	0.009
**Lactase activity ^2^**	1.56 ± 0.11 ^ab^	0.816 ± 0.09^c^	1.73 ± 0.12 ^a^	1.33 ± 0.11 ^abc^	1.16 ± 0.12 ^bc^	0.001
**Cecum**						
**Tissue ^1^**	0.201 ± 0.002 ^a^	0.182 ± 0.003 ^ab^	0.192 ± 0.003 ^ab^	0.197 ± 0.001 ^a^	0.173 ± 0.002 ^b^	0.010
**Content ^1^**	0.586 ± 0.012 ^a^	0.436 ± 0.020 ^ab^	0.495 ± 0.019 ^ab^	0.509 ± 0.025 ^ab^	0.386 ± 0.021 ^b^	0.016
**DM (%)**	24.0 ± 0.41 ^ab^	24.5 ± 0.46 ^ab^	22.5 ± 0.39 ^b^	23.7 ± 0.40 ^ab^	26.1 ± 0.41 ^a^	0.010
**NH_3_ (mg/g)**	0.227 ± 0.007 ^ab^	0.254 ± 0.009 ^a^	0.213 ± 0.008 ^ab^	0.188 ± 0.009 ^b^	0.210 ± 0.005 ^ab^	0.016
**Content pH**	7.21 ± 0.04	7.34 ± 0.05	7.34 ± 0.04	7.41 ± 0.04	7.44 ± 0.05	0.197
**Colon**						
**Tissue ^1^**	0.325 ± 0.018	0.279 ± 0.013	0.292 ± 0.012	0.290 ± 0.013	0.285 ± 0.010	0.055
**Content ^1^**	0.205 ± 0.012	0.222 ± 0.011	0.230 ± 0.015	0.238 ± 0.015	0.195 ± 0.018	0.502
**Content pH**	7.42 ± 0.05 ^ab^	7.50 ± 0.04 ^ab^	7.24 ± 0.08 ^b^	7.65 ± 0.05 ^a^	7.48 ± 0.04 ^ab^	0.049

Diet: D_CS_—control standard diet, D_CF_—control high-fat low-fiber diet, D_RW_—diet enriched with freeze-dried raw cocoa bean extract, D_RT_—diet enriched with freeze-dried roasted cocoa bean extract, D_CT_—diet enriched with freeze-dried monomeric flavan-3-ols fraction; SEM: standard error of the mean, SD for all rats divided by square root of an overall number of rats, *n* = 40; Mean values not sharing the same superscript letters within a row are significantly different at *p*
≤ 0.05; ^1^ g/100 g BW, ^2^ µmol/min/g protein.

**Table 2 molecules-24-00825-t002:** Activity of the selected bacterial enzymes in the rats’ feces during the experiment. Data are expressed as mean ± SEM, *n* = 8.

	Diet					*p*-Value
	D_CS_	D_CF_	D_RW_	D_RT_	D_CT_	
**β-glucosidase**						
day 0	39.0 ± 1.4	40.6 ± 1.5	35.1 ± 1.6	37.3 ± 1.7	38.6 ± 1.4	0.499
day 1	40.9 ± 2.7 ^c^	111.0 ± 3.2 ^a^	101.0 ± 4.5 ^a^	71.7 ± 3.9 ^b^	58.2 ± 2.7 ^bc^	<0.001
day 4	19.0 ± 1.1 ^bc^	24.0 ± 1.3 ^ab^	20.8 ± 1.4 ^ab^	26.7 ± 1.4 ^a^	13.6 ± 1.3 ^c^	<0.001
day 8	10.5 ± 0.7 ^b^	12.7 ± 0.5 ^ab^	14.7 ± 0.8 ^ab^	16.5 ± 0.6 ^a^	13.1 ± 0.8 ^ab^	0.015
day 11	7.19 ± 0.78 ^b^	9.02 ± 0.83 ^b^	14.10 ± 0.86 ^a^	14.20 ± 0.79 ^a^	10.40 ± 0.71 ^ab^	0.009
**β-galactosidase**						
day 0	168.0 ± 1.9	164.0 ± 1.8	161.0 ± 1.7	168.0 ± 1.6	168.0 ± 1.8	0.326
day 1	213.0 ± 5.0 ^d^	528.0 ± 6.0 ^a^	400.0 ± 9.0 ^abc^	480.0 ± 7.0 ^ab^	290.0 ± 8.0 ^cd^	<0.001
day 4	134 ± 9.0 ^c^	238.0 ± 7.0 ^a^	206.0 ± 8.0 ^ab^	164.0 ± 6.0 ^bc^	170.0 ± 8.0 ^bc^	<0.001
day 8	81.6 ± 7.4 ^b^	130.0 ± 8.2 ^a^	170.0 ± 7.9 ^a^	140.0 ± 7.5 ^a^	157.0 ± 7.1 ^a^	0.001
day 11	84.9 ± 7.4 ^b^	131.0 ± 7.1 ^a^	146.0 ± 6.8 ^a^	134.0 ± 7.2 ^a^	151.0 ± 7.5 ^a^	0.008
**β-glucuronidase**						
day 0	97.0 ± 2.5	100.0 ± 1.8	93.9 ± 2.7	94.2 ± 1.9	94.4 ± 2.5	0.615
day 1	157.0 ± 5.2 ^c^	271.0 ± 6.9 ^a^	253.0 ± 7.1 ^a^	226.0 ± 4.8 ^b^	173.0 ± 8.1 ^c^	<0.001
day 4	48.2.0 ± 3.9 ^c^	70.4 ± 4.2 ^b^	114.0 ± 3.8 ^a^	77.6 ± 4.5 ^b^	46.5 ± 3.7 ^c^	<0.001
day 8	53.1 ± 2.6 ^bc^	60.8 ± 3.9 ^bc^	86.7 ± 3.2 ^a^	68.0 ± 3.5 ^ab^	46.1 ± 3.8 ^c^	<0.001
day 11	34.3 ± 3.1 ^b^	39.9 ± 2.9 ^b^	60.2 ± 3.1 ^a^	60.8 ± 3.2 ^a^	47.4 ± 2.8 ^ab^	0.010

Diet: D_CS_—control standard diet, D_CF_—control high-fat low-fiber diet, D_RW_—diet enriched with freeze-dried raw cocoa bean extract, D_RT_—diet enriched with freeze-dried roasted cocoa bean extract, D_CT_—diet enriched with freeze-dried monomeric flavan-3-ols fraction; SEM: standard error of the mean, SD for all rats divided by square root of an overall number of rats, *n* = 40; Mean values not sharing the same superscript letters within a row are significantly different at *p*
≤ 0.05.

**Table 3 molecules-24-00825-t003:** Activity of the selected bacterial enzymes in the rats’ cecum content at the end of the experiment. Data are expressed as mean ± SEM, *n* = 8.

	Diet					*p*-Value
	D_CS_	D_CF_	D_RW_	D_RT_	D_CT_	
**α-glucosidase**						
Extracellular ^1^	20.4 ± 0.5 ^a^	17.0 ± 0.7 ^ab^	17.8 ± 0.5 ^ab^	13.6 ± 0.6 ^b^	17.3 ± 0.7 ^ab^	0.007
Intracellular ^1^	4.12 ± 0.8 ^b^	1.61 ± 0.9 ^b^	12.30 ± 0.9 ^a^	9.16 ± 0.7 ^a^	2.22 ± 0.8 ^b^	<0.001
total ^1^	24.6 ± 0.9 ^ab^	18.6 ± 1.1 ^b^	30.1 ± 0.9 ^a^	22.8 ± 0.8 ^b^	19.5 ± 1.2 ^b^	0.002
release degree ^2^	84.5 ± 1.5 ^a^	91.0 ± 0.9 ^a^	61.4 ± 1.9 ^b^	61.0 ± 0.8 ^b^	88.6 ± 1.8 ^a^	<0.001
**β-glucosidase**						
Extracellular ^1^	3.63 ± 0.20 ^a^	3.13 ± 0.12 ^a^	2.76 ± 0.13 ^ab^	2.50 ± 0.19 ^ab^	1.84 ± 0.15 ^b^	0.007
Intracellular ^1^	3.02 ± 0.21 ^a^	0.61 ± 0.31 ^b^	3.68 ± 0.26 ^a^	4.08 ± 0.17 ^a^	1.02 ± 0.32 ^b^	<0.001
Total ^1^	6.65 ± 0.36 ^a^	3.74 ± 0.39 ^b^	6.44 ± 0.39 ^a^	6.58 ± 0.35 ^a^	2.85 ± 0.41 ^b^	<0.001
release degree ^2^	56.1 ± 2.7 ^b^	81.9 ± 3.1 ^a^	42.3 ± 3.6 ^c^	39.0 ± 2.7 ^c^	65.2 ± 3.1 ^b^	<0.001
**α-galactosidase**						
Extracellular ^1^	10.80 ± 0.71	11.00 ± 0.61	8.54 ± 0.54	11.30 ± 0.59	7.72 ± 0.61	0.134
Intracellular ^1^	7.90 ± 1.1 ^b^	5.98 ± 0.9 ^b^	14.40 ± 0.9 ^a^	19.40 ± 1.1 ^a^	5.70 ± 1.0 ^b^	<0.001
Total ^1^	18.7 ± 1.5 ^bc^	17.0 ± 1.7 ^bc^	22.9 ± 1.4 ^b^	30.7 ± 1.5 ^a^	13.4 ± 1.6 ^c^	<0.001
release degree ^2^	57.7 ± 1.9 ^a^	66.5 ± 2.7 ^a^	38.1 ± 1.9 ^b^	34.5 ± 2.5 ^b^	59.2 ± 1.9 ^a^	<0.001
**β-galactosidase**						
Extracellular ^1^	44.4 ± 2.5 ^b^	71.0 ± 2.9 ^a^	60.9 ± 2.7 ^a^	40.6 ± 3.1 ^b^	62.0 ± 2.9 ^a^	<0.001
Intracellular ^1^	20.7 ± 1.7 ^ab^	21.3 ± 1.5 ^ab^	29.2 ± 1.6 ^a^	25.1 ± 1.7 ^ab^	18.0 ± 1.6 ^b^	0.041
total ^1^	65.1 ± 3.9 ^b^	92.3 ± 4.1 ^a^	90.1 ± 3.9 ^a^	65.7 ± 3.5 ^b^	80.0 ± 3.4 ^ab^	0.026
release degree ^2^	69.3 ± 1.5 ^ab^	77.8 ± 1.7 ^a^	67.3 ± 1.6 ^b^	62.6 ± 1.5 ^b^	77.2 ± 1.7 ^a^	0.003
**β-glucuronidase**						
Extracellular ^1^	21.4 ± 0.9	22.0 ± 1.1	20.0 ± 0.9	17.2 ± 1.2	18.7 ± 0.9	0.240
Intracellular ^1^	12.00 ± 0.76	7.22 ± 0.81	11.20 ± 0.84	10.50 ± 0.86	6.97 ± 0.89	0.109
Total ^1^	33.3 ± 1.4	29.2 ± 1.5	31.2 ± 1.6	27.6 ± 1.7	25.7 ± 1.4	0.155
release degree ^2^	64.8 ± 1.9 ^ab^	78.1 ± 2.4 ^a^	64.6 ± 2.5 ^ab^	61.7 ± 1.7 ^ab^	73.5 ± 1.9 ^ab^	0.046

Diet: D_CS_—control standard diet, D_CF_—control high-fat low-fiber diet, D_RW_—diet enriched with freeze-dried raw cocoa bean extract, D_RT_—diet enriched with freeze-dried roasted cocoa bean extract, D_CT_—diet enriched with freeze-dried monomeric flavan-3-ols fraction; SEM: standard error of the mean, SD for all rats divided by square root of an overall number of rats, *n* = 40; Mean values not sharing the same superscript letters within a row are significantly different at *p*
≤ 0.05; ^1^ µmol/h/g content; ^2^ extracellular as of total activity.

**Table 4 molecules-24-00825-t004:** Volatile fatty acids in the rats’ cecum content. Data are expressed as mean ± SEM, *n* = 8.

	Diet					*p*-Value
	D_CS_	D_CF_	D_RW_	D_RT_	D_CT_	
**VFA (µmol/g content)**						
acetic	73.1 ± 1.9	71.4 ± 2.2	86.7 ± 1.9	75.9 ± 2.3	77.5 ± 2.1	0.052
propionic	15.8 ± 0.5 ^ab^	13.8 ± 0.6 ^b^	19.2 ± 0.7 ^a^	19.0 ± 0.5 ^a^	16.5 ± 0.6 ^ab^	0.012
iso-butyric	1.84 ± 0.07 ^a^	1.37 ± 0.06 ^ab^	1.53 ± 0.07 ^ab^	1.21 ± 0.05 ^b^	1.40 ± 0.08 ^ab^	0.014
butyric	10.70 ± 0.51 ^a^	8.70 ± 0.48 ^ab^	7.52 ± 0.51 ^b^	6.16 ± 0.46 ^b^	7.31 ± 0.53 ^b^	0.005
iso-valeric	1.82 ± 0.06 ^a^	1.26 ± 0.04 ^b^	1.34 ± 0.05 ^b^	1.06 ± 0.05 ^b^	1.18 ± 0.07 ^b^	<0.001
valeric	3.23 ± 0.25	1.78 ± 0.19	1.91 ± 0.21	1.92 ± 0.23	1.75 ± 0.21	0.062
Sum of iso-butyric, iso-valeric and valeric acid	6.89 ± 0.21 ^a^	4.42 ± 0.27 ^b^	4.77 ± 0.24 ^b^	4.20 ± 0.28 ^b^	4.33 ± 0.23 ^b^	0.002
VFA total	107.0 ± 2.6	98.3 ± 2.9	118.0 ± 3.1	105.0 ± 3.2	106.0 ± 3.1	0.071
**VFA pool (µmol/100 g BW)**						
acetic	43.1 ± 1.5	31.6 ± 1.9	42.4 ± 2.2	38.8 ± 2.3	29.9 ± 1.8	0.069
propionic	9.45 ± 0.46 ^ab^	6.21 ± 0.51 ^b^	9.40 ± 0.54 ^ab^	9.65 ± 0.53 ^a^	6.30 ± 0.55 ^ab^	0.049
butyric	6.41 ± 0.32 ^a^	3.90 ± 0.31 ^b^	3.66 ± 0.29 ^b^	3.14 ± 0.38 ^b^	2.73 ± 0.36 ^b^	0.002
Sum of iso-butyric, iso-valeric and valeric acid	4.01 ± 0.21 ^a^	1.97 ± 0.19 ^b^	2.34 ± 0.18 ^b^	2.18 ± 0.22 ^b^	1.69 ± 0.20 ^b^	<0.001
VFA total	63.0 ± 2.5 ^a^	43.7 ± 3.4 ^ab^	57.8 ± 2.8 ^ab^	53.8 ± 2.7 ^ab^	40.6 ± 2.9 ^b^	0.032
**VFA profile (% total)**						
acetic	68.8 ± 0.5 ^b^	73.3 ± 0.5 ^a^	73.4 ± 0.6 ^a^	72.1 ± 0.6 ^a^	73.1 ± 0.5 ^a^	0.011
propionic	15.0 ± 0.5 ^ab^	13.2 ± 0.4 ^b^	16.3 ± 0.6 ^ab^	18.0 ± 0.5 ^a^	15.9 ± 0.4 ^ab^	0.005
butyric	9.88 ± 0.34 ^a^	8.90 ± 0.39 ^a^	6.31 ± 0.41 ^b^	5.96 ± 0.36 ^b^	6.86 ± 0.38 ^b^	0.001

Diet: D_CS_—control standard diet, D_CF_—control high-fat low-fiber diet, D_RW_—diet enriched with freeze-dried raw cocoa bean extract, D_RT_—diet enriched with freeze-dried roasted cocoa bean extract, D_CT_—diet enriched with freeze-dried monomeric flavan-3-ols fraction; BW—body weight, VFA—Volatile Fatty Acids; SEM: standard error of the mean, SD for all rats divided by square root of an overall number of rats, *n* = 40; Mean values not sharing the same superscript letters within a row are significantly different at *p*
≤ 0.05.

**Table 5 molecules-24-00825-t005:** Hematological parameters in rats. Data are expressed as mean ± SEM, *n* = 8.

	Diet					*p*-Value
	D_CS_	D_CF_	D_RW_	D_RT_	D_CT_	
WBC (10^3^/μL)	5.70 ± 0.19 ^ab^	6.11 ± 0.21 ^ab^	6.54 ± 0.20 ^a^	6.23 ± 0.19 ^ab^	4.97 ± 0.22 ^b^	0.032
LYM (10^3^/μL)	4.67 ± 0.21 ^ab^	4.97 ± 0.16 ^ab^	5.48 ± 0.17 ^a^	5.06 ± 0.19 ^ab^	4.10 ± 0.18 ^b^	0.040
MID (10^3^/μL)	0.361 ± 0.018 ^a^	0.347 ± 0.014 ^a^	0.171 ± 0.019 ^b^	0.256 ± 0.025 ^ab^	0.242 ± 0.021 ^ab^	0.015
GRA (10^3^/μL)	0.665 ± 0.049	0.807 ± 0.051	0.890 ± 0.055	0.917 ± 0.048	0.681 ± 0.051	0.108
LYM (%)	82.1 ± 0.9	81.1 ± 0.8	82.8 ± 0.9	81.6 ± 0.7	82.3 ± 1.1	0.626
MID (%)	6.18 ± 0.32 ^a^	5.68 ± 0.38 ^ab^	2.76 ± 0.34 ^c^	3.86 ± 0.38 ^bc^	4.74 ± 0.36 ^abc^	0.002
GRA (%)	11.8 ± 0.8	13.3 ± 0.9	14.5 ± 0.7	14.6 ± 0.8	13.0 ± 0.9	0.347
RBC (10^3^/μL)	7.47 ± 0.05	7.59 ± 0.07	7.76 ± 0.08	7.83 ± 0.05	7.76 ± 0.03	0.054
HGB (g/dL)	13.8 ± 0.1 ^c^	14.1 ± 0.1 ^abc^	14.6 ± 0.1 ^a^	14.4 ± 0.1 ^ab^	14.0 ± 0.1 ^bc^	0.010
HCT (%)	40.2 ± 0.3 ^b^	40.8 ± 0.2 ^ab^	42.6 ± 0.4 ^a^	42.6 ± 0.3 ^a^	41.1 ± 0.2 ^ab^	0.020
MCV (fL)	53.7 ± 0.2 ^ab^	53.8 ± 0.3 ^ab^	54.9 ± 0.2 ^a^	54.4 ± 0.3 ^ab^	52.9 ± 0.1 ^b^	0.035
MCH (pg)	18.4 ± 0.1	18.6 ± 0.2	18.8 ± 0.1	18.5 ± 0.1	18.0 ± 0.2	0.121
MCHC (g/dL)	34.3 ± 0.2	34.6 ± 0.1	34.3 ± 0.1	33.8 ± 0.2	34.0 ± 0.1	0.121
RDWc (%)	17.5 ± 0.1 ^bc^	17.3 ± 0.1 ^c^	17.6 ± 0.1 ^abc^	18.0 ± 0.1 ^a^	17.8 ± 0.1 ^ab^	0.004
PLT (10^3^/μL)	482 ± 4	468 ± 6	520 ± 8	514 ± 10	485 ± 9	0.087
PCT (%)	0.372 ± 0.010	0.367 ± 0.005	0.400 ± 0.006	0.381 ± 0.007	0.371 ± 0.009	0.227
MPV (fL)	7.73 ± 0.07 ^ab^	7.83 ± 0.06 ^a^	7.33 ± 0.04 ^c^	7.41 ± 0.05 ^bc^	7.68 ± 0.06 ^abc^	0.013
PDWc (%)	33.8 ± 0.2 ^ab^	35.0 ± 0.1 ^a^	33.2 ± 0.3 ^b^	33.8 ± 0.2 ^ab^	35.0 ± 0.1 ^a^	0.007

Diet: D_CS_—control standard diet, D_CF_—control high-fat low-fiber diet, D_RW_—diet enriched with freeze-dried raw cocoa bean extract, D_RT_—diet enriched with freeze-dried roasted cocoa bean extract, D_CT_—diet enriched with freeze-dried monomeric flavan-3-ols fraction; WBC: white blood cells (norm 2.1–19.5 × 10^3^/μL), LYM: lymphocytes (norm 2–14.1 × 10^3^/μL; 55–97%), MID: intermediate forms between lymphocytes and neutrophils (norm 0–0.98; 0–5%), GRA: granulocytes (norm 0.1–5.4 × 10^3^/μL; 2–31%), RBC: red blood cells (norm 5.3–10 × 10^6^/μL), HGB: hemoglobin concentration (norm 14–18 g/dL), HCT: hematocrit (norm 35–52%), MCV: mean corpuscular volume (norm 50–62 fL), MCH: mean corpuscular hemoglobin (norm 16–23 pg), MCHC: mean corpuscular hemoglobin concentration (norm 31–40 g/dL), RDWc: erythrocyte volumetric variability index, PLT: platelets (500–1370 × 10^3^/μL), PCT: platelet hematocrit, MPV: mean platelet volume; PDWc: platelet volumetric variability index; SEM: standard error of the mean, SD for all rats divided by square root of an overall number of rats, *n* = 40; Mean values not sharing the same superscript letters within a row are significantly different at *p*
≤ 0.05.

**Table 6 molecules-24-00825-t006:** Composition of the experimental diets.

(%)	Diet ^1^				
	D_CS_	D_CF_	D_RW_	D_RT_	D_CT_
Casein	20.0	20.0	20.0	20.0	20.0
DL-methionine	0.3	0.3	0.3	0.3	0.3
Rapeseed oil	7.0	7.0	7.0	7.0	7.0
Palm oil	-	14.0	14.0	14.0	14.0
Cellulose	5.0	2.0	2.0	2.0	2.0
Sucrose	10.0	10.0	10.0	10.0	10.0
Mineral blend	3.5	3.5	3.5	3.5	3.5
Vitamin blend	1.0	1.0	1.0	1.0	1.0
Choline chloride	0.2	0.2	0.2	0.2	0.2
Raw cocoa extract	-	-	2.25	-	-
Roasted cocoa extract	-	-	-	2.45	-
Monomeric flavan-3-ols fraction	-	-	-	-	0.114
Corn starch	53.0	42.0	39.75	39.55	41.886

^1^ Diet: D_CS_—control standard diet, D_CF_—control high-fat diet, D_RW_—diet enriched with freeze-dried raw cocoa bean extract, D_RT_—diet enriched with freeze-dried roasted cocoa bean extract, D_CT_—diet enriched with freeze-dried monomeric flavan-3-ols fraction.

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
