# Peer review of "Influence of Diet Enriched with Cocoa Bean Extracts on Physiological Indices of Laboratory Rats"

_molecules, 2019, doi:10.3390/molecules24050825_

Round 1

Reviewer 1 Report

Review of Manuscript ID: molecules-440501

Title: Influence of diet enriched with cocoa bean extracts on physiological indices of laboratory rats

Authors: Dorota Zyzelewicz *, Malgorzata Bojczuk, Grazyna Budryn, Zenon Zdunczyk, Jerzy Juskiewicz, Adam Jurgonski, Joanna Oracz

General comment: the study reports the potential protective effects of cocoa on rat intestinal health against a high fat diet. The study focus on a number of essential small intestine and colonic bacterial enzymes, several volatile fatty acids and a plethora of blood parameters in rats submitted to standard diet, high-fat rich diet and the same fat diet supplemented with either raw cocoa extract, roasted cocoa extract or an extract enriched in monomeric fraction of catechins. Although some particular changes observed in high-fat diet rats were reverted by coco extracts, most parameters evaluated did not change in rats submitted to the fat diet, thus, the potential protective effect of cocoa remained unrevealed. As stated by the authors in the conclusions, the experimental period was too short and insufficient to provoke measurable physiological and biochemical alterations. Some specific comments are detailed below:

Specific comments:

Lines 68-85; the recovery of decreased lactase activity in Dcf by raw cocoa extract (Drw) should also be described.

Lines 143-147; it should not be stated that there was a clearly higher level of acetate in Drw when there are not significant differences as compared to the rest of groups. Similarly, it should not be reported that total production (pool) of VFAs in the cecum content was the highest in Dcs group when the only group with a statistically significant decrease was Dct. The paragraph should be rewritten in a more appropriate manner.

Lines 149-151; the meaning of the sentence is not clear.

Tables 3-5; since the fat diet treatment failed to modify any single hematological parameter and most of bacterial enzymes and volatile fatty acids in cecum and feces also remained unaltered, this does not seem to be an appropriate model to show the potential chemo-protective effect of cocoa extracts or specific favanols.

Lines 200-226; only polyphenols, mostly flavonoids, are described as components of the cocoa bean extract, but other important phytochemicals such as theobromine (more abundant than most flavanols) should also be included in the chemical characterization of the extracts.

Table 6; addition of distinct amounts of the three cocoa supplements to the diet should be explained. It should be clearly stated why 2.25 % of raw cocoa, 2.45 % of toasted cocoa and 0.11 % of monomeric flavanols were added to the diets.

Lines 320-323; no redox parameter whatsoever is described in the manuscript.

Lines 340-342; the sentence ratifies that the experimental model was insufficient to confirm most of the purported beneficial effects of cocoa on gastrointestinal function. The authors should clearly remark this limitation in the objective of the study by asserting that it is focused on the early stages of a high fat diet intake.

Author Response

Dear Reviewer,

I express sincere thanks to Reviewer for taking the trouble to analyze the article and drawing up a thorough review. I believe that this will make our publication more valuable. We obeyed all remarks made by Reviewer. Nevertheless, below are provided detailed responses to comments.

Reviewer’s comment:

Lines 68-85; the recovery of decreased lactase activity in DcF by raw cocoa extract (DRw) should also be described.

Answer:

Thank you for your comment. The sentence has been corrected.

Reviewer’s comment:

Lines 143-147; it should not be stated that there was a clearly higher level of acetate in DRw when there are not significant differences as compared to the rest of groups. Similarly, it should not be reported that total production (pool) of VFAs in the cecum content was the highest in Dcs group when the only group with a statistically significant decrease was DcT. The paragraph should be rewritten in a more appropriate manner.

Answer:

Thank you for your valuable comment. The sentence has been corrected:

The total production (pool) of VFAs in the cecum content was only slightly different between diets, however, a statistically significant difference (p≤0.05) was noted only in the case of a diet containing monomeric flavan3-ols fraction.

Reviewer’s comment:

Lines 149-151; the meaning of the sentence is not clear.

Answer:

The sentence has been changed. Thank you very much.

Reviewer’s comment:

Tables 3-5; since the fat diet treatment failed to modify any single hematological parameter and most of bacterial enzymes and volatile fatty acids in cecum and feces also remained unaltered, this does not seem to be an appropriate model to show the potential chemo-protective effect of cocoa extracts or specific favanols.

Answer:

Thank you for your comment. We agree that the high-fat low-fiber diet and also relatively short period of feeding (4 weeks) appeared not to be optimal to induce disorders in a rat organism, especially when considering hematological parameters. Thus, in the conclusion section, the following sentences: “the results of some experiments described in this paper indicate that a relatively short period of diet administration, namely 4 weeks, was not fully sufficient to clearly confirm course of the changes observed within tested groups. Therefore, the material such as cocoa beans is recommended for the further analysis.”

have been changed to:

“Nevertheless, a relatively short period of diet administration, namely 4 weeks, was not fully sufficient to clearly confirm course of the changes observed within tested groups and the high-fat low-fiber diet used in this study was not able to disturb hematological parameters. Therefore, the potential chemoprotective effect of cocoa extracts could not be observed and the material is recommended for further analysis.”

 Reviewer’s comment:

Lines 200-226; only polyphenols, mostly flavonoids, are described as components of the cocoa bean extract, but other important phytochemicals such as theobromine (more abundant than most flavanols) should also be included in the chemical characterization of the extracts.

Answer:

Information on the content of theobromine and caffeine in the extracts (p. 7, lines 223-225) and preparation monomeric flavan-3-ols is given in the text (p. 7, lines 246-248)

Reviewer’s comment:

Table 6; addition of distinct amounts of the three cocoa supplements to the diet should be explained. It should be clearly stated why 2.25 % of raw cocoa, 2.45 % of toasted cocoa and 0.11 % of monomeric flavanols were added to the diets.

Answer:

The explanation of various doses of extracts and monomeric flavan-3-ols is presented in the text in section 3.3.1 (p. 8,  lines 264-268):

Previous studies have demonstrated that in very high doses, flavonoids can act as pro‐oxidant [41]. Therefore, due to the fact that the fraction of monomeric flavan-3-ols have an approximately 20-fold higher polyphenols content than extracts of raw and roasted cocoa beans, various additives of the tested preparations were applied to the diets so as to obtain a similar level of total polyphenols in the diet.

Reviewer’s comment:

Lines 320-323; no redox parameter whatsoever is described in the manuscript.

Answer:

Indeed, this paragraph was mistakenly left from the first version of the manuscript. Thus, the paragraph has been removed.

Reviewer’s comment:

Lines 340-342; the sentence ratifies that the experimental model was insufficient to confirm most of the purported beneficial effects of cocoa on gastrointestinal function. The authors should clearly remark this limitation in the objective of the study by asserting that it is focused on the early stages of a high fat diet intake.

Answer:

Thank you for your valuable comment. As suggested, the last paragraph of the Introduction clearly states that research is focused on the early stages of a high fat diet intake.

We thank once more for all the comments. We hope that changes applied in the article are consistent with the suggestions made by Reviewer and will be satisfying and approved. Thank you again

On behalf of the research team

Dorota Żyżelewicz

Reviewer 2 Report

This is a very interesting study that shows the effect of cocoa bean extracts on the physiological indices of laboratory rats.

Would the authors please clarify the sentence in lines 75-76 of the manuscript. In addition at the end of the sentence in line 81 should one be changed to group?

Author Response

Dear Reviewer,

I express sincere thanks to Reviewer for taking the trouble to analyze the article and drawing up a thorough review. We obeyed remark made by the Reviewer. Nevertheless, below is provided detailed response to comments.

Reviewer’s comments:

Would the authors please clarify the sentence in lines 75-76 of the manuscript. In addition at the end of the sentence in line 81 should one be changed to group?

Answer:

Thank you for your comment. The sentences have been corrected (p. 2, lines 76-78).

On behalf of the research team

Dorota Żyżelewicz

Round 2

Reviewer 1 Report

The authors have conveniently addressed all my comments and queries, therefore, I recommend to accept the revised version of the manuscript for publication at Molecules

Author Response

Thank you for accepting changes in the publication.